# Learning to Reason About Code Insecurity: Composite-Reinforcement Fine-Tuning for Cognitive Alignment

## Abstract

Automated vulnerability analysis increasingly relies on language models, yet even strong LLMs exhibit unstable security reasoning: they either over-flag benign code or miss critical flaws, particularly under cross-language shifts. We present **ARGO**—*Composite-Reinforcement Fine-Tuning for Cognitive Alignment*—a label-efficient training framework that explicitly optimizes a composite reward combining (i) *label-based decision scoring* via a strictly proper scoring rule on predicted probabilities, (ii) *explanation grounding and consistency* through structure- and code-referencing heuristics that do not use Common Weakness Enumeration (CWE) labels or definitions, and (iii) *output-format coherence* through a strict schema validator. This moves the objective from bare classification toward deliberative, auditable analysis while explicitly acknowledging and isolating the supervised component in the reward. We cast each example as a short two-phase episode: first, the policy produces an explanation; then it deterministically emits a calibrated probability through a regression head. The binary decision is deterministically derived from the probability at inference (thresholding) rather than being sampled as a separate action. Policy updates are stabilized via batch-level affinity-weighted neighborhood smoothing over deterministic encoding and a KL trust term to a reference policy. Across BigVul, DiverseVul, and CleanVul, ARGO consistently improves macro-F1 over strong baselines (e.g., up to 0.71 in-distribution; substantial gains under cross-language transfer). Compared to standard supervised fine-tuning, ARGO reduces catastrophic bias toward predicting the vulnerable class and improves recognition of benign code without relying on CWE supervision.

## 1 Introduction

Vulnerability detection is decision-critical: beyond raw accuracy, *reasoning quality* and *auditability* govern trust. In practice, developers and auditors require *grounded explanations* that reference concrete code artifacts (variables, call sites, bounds, sanitization logic) rather than opaque labels. Yet recent studies observe that LLMs remain *unstable* on memory-related weaknesses and domain idioms: they oscillate between over-cautious (over-flagging) and permissive (missing critical flaws) behaviors, and their rationales often fail to align with code structure despite prompt engineering. We target compact instruction-tuned backbones and ask: can we *align* them toward **principled, grounded reasoning** using a training objective that *explicitly rewards* both calibration and evidential explanations? Our answer is ARGO (as shown in Figure 1): a composite-reinforcement fine-tuning scheme that (1) retains a supervised *proper scoring* term for calibrated probabilities; (2) adds *unsupervised* explanation/format signals computed purely from $(x, e, p)$; and (3) uses a *two-phase* interface (explain → emit probability) with KL trust regularization. The method is label-efficient, avoids ambiguous "self-supervised" claims, and—empirically—reduces catastrophic biases while improving benign recognition.

We adopt a defensive threat model: detectors should (i) surface plausible weaknesses with calibrated uncertainty, (ii) refrain from overwhelming analysts with false positives on benign idioms, and (iii) produce *code-grounded* explanations that accelerate audits. Operational desiderata include: calibration, format reliability, cross-language transferability, reproducibility, and safety. Through-

out, **ARGO** denotes *Composite-Reinforcement Fine-Tuning for Cognitive Alignment* (the expansion used in the title). Earlier drafts used an alternative descriptive phrase; we standardize on the title expansion to avoid ambiguity. Detection systems are most valuable when they encourage the same habits that skilled reviewers practice: read code with attention to naming and control flow, articulate a concise hypothesis for how data move, and only then commit to a decision with a quantifiable degree of confidence. The two-phase interface of ARGO mirrors this routine. Phase 1 fixes attention on *what the model thinks matters* by requiring explicit references to identifiers and spans that appear in the snippet; Phase 2 converts this narrative into a single calibrated probability via a deterministic regression head, avoiding additional sampling noise and making gradients straightforward for proper scoring.

Calibration is necessary but not sufficient: an ungrounded explanation paired with a well-calibrated probability offers little guidance during triage, while a grounded but overconfident explanation can flood issue trackers with false alarms. The composite reward therefore places a proper scoring rule in direct tension with unlabelled signals that promote concrete references and format reliability. The combined pressure discourages pathological shortcuts such as echoing dataset-specific phrases, amplifying spurious lexical co-occurrences, or defaulting to the vulnerable class when the snippet merely *resembles* a known template. Because the explanation reward is computed without Common Weakness Enumeration (CWE) supervision and without external retrieval, it remains robust to gaps in curated taxonomies and avoids tying generalization to retrieval quality. We restrict our scope to short code-centered episodes and do not claim capability for exploit synthesis, patch generation, or formal verification. The algorithm does not depend on private or proprietary vulnerability corpora and is designed to operate with compact backbones under modest compute budgets. We intentionally avoid training-time exposure to CWE definitions or labels; any semantic alignment to CWE categories is an *evaluation-only* diagnostic with decoy controls. The objective of ARGO is not to discover entirely new classes of defects but to make predictions about existing classes more dependable, transparent, and transferable across languages.

In operational settings, analysts frequently maintain risk budgets and escalation policies. ARGO produces calibrated probabilities that can be thresholded to match such policies, while the structured rationale fields (`Issue/Evidence/Mitigation`) provide compact artifacts for code reviews and audit trails. Because the decision is deterministically derived from the probability at inference time, downstream systems can sweep operating points without retraining, which is preferable in organizations that must document when and why a change in sensitivity occurred.

**Contributions.** (1) A **composite-reinforcement** objective combining supervised decision scoring with unsupervised grounding/format signals—*explicitly* stated to avoid terminology ambiguity. (2) A **two-phase** formulation that matches the task and stabilizes training with KL regularization and affinity smoothing, with a deterministic regression head for calibrated probabilities. (3) A **reproducible evaluation protocol** featuring cross-dataset near-duplicate removal, ablations, multi-seed statistics, and evaluation-only CWE analyses with negative controls.

## 2 RELATED WORK

The landscape of automated vulnerability detection has undergone substantial transformation with the emergence of deep learning architectures and large language models, yet fundamental challenges persist in achieving reliable, interpretable, and transferable security analysis. Recent advances have explored diverse representations and learning paradigms, from graph-based encodings that capture program structure to attention mechanisms that identify critical code patterns, though these approaches often operate as opaque classifiers that provide limited insight into their reasoning processes Fan et al. (2020); Chen et al. (2023); Li et al. (2024). The introduction of transformer-based architectures has enabled more sophisticated pattern recognition across longer code sequences, with models like CodeBERT and GraphCodeBERT demonstrating improved performance on standard benchmarks, yet these gains frequently fail to translate into robust detection capabilities when confronted with real-world code that deviates from training distributions Wang et al. (2025); Chen et al. (2025).

Contemporary research has increasingly recognized that pure accuracy metrics inadequately capture the requirements of practical vulnerability analysis, where false positives impose substantial triage

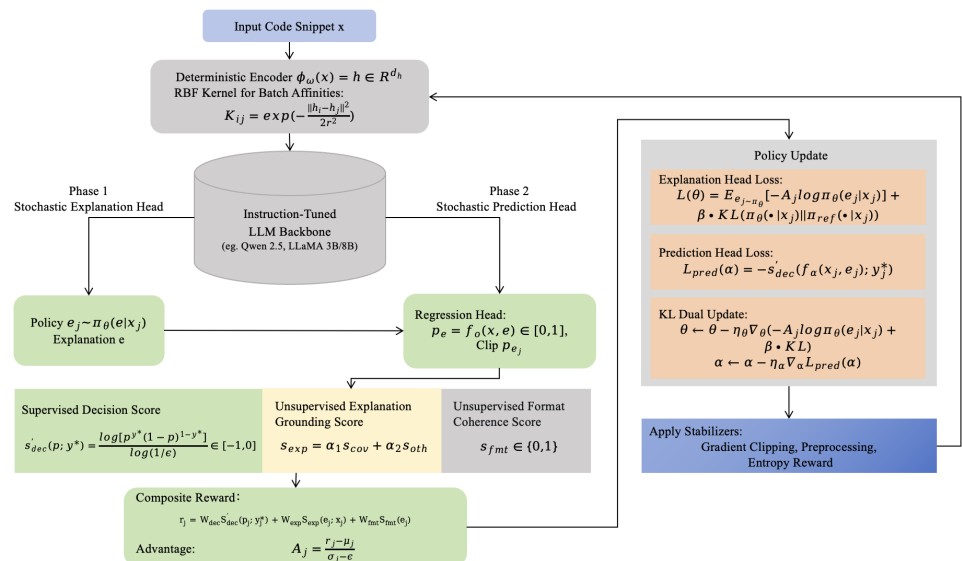

Figure 1: ARGO Model Architecture: Two-Phase Episode with Composite Reinforcement.

costs and ungrounded predictions undermine developer trust. The work of Zhang et al. Zhang et al. (2025) introduced hierarchical attention mechanisms that attempt to localize vulnerability-relevant code regions, though their approach still relies heavily on implicit pattern matching rather than explicit reasoning about program semantics. Similarly, Chang and colleagues Chang et al. (2025) proposed structure-aware embeddings that encode control flow and data dependencies, achieving notable improvements on memory corruption vulnerabilities, yet their method remains susceptible to spurious correlations between lexical features and vulnerability labels. The GTVD framework by He et al. He et al. (2025) advances graph-based representations through temporal modeling of code evolution, capturing how vulnerabilities emerge through software maintenance, though this approach requires extensive historical data that may not be available for newly developed systems.

The application of large language models to vulnerability detection has opened new avenues for incorporating semantic understanding and generating human-readable explanations, though significant gaps remain between model capabilities and operational requirements. Recent investigations by Ding et al. Ding et al. (2025) systematically evaluated GPT-4 and similar models on diverse vulnerability types, revealing persistent instabilities when analyzing memory management errors and domain-specific idioms that deviate from common training patterns. Zhou and collaborators Zhou et al. (2025) conducted extensive prompt engineering experiments to elicit more reliable security reasoning from LLMs, discovering that even carefully crafted prompts fail to prevent oscillation between over-cautious false positives and dangerous false negatives. The comprehensive study by Pan et al. Pan et al. (2025) examined cross-language transfer capabilities of modern language models, documenting substantial performance degradation when models trained on one programming language are applied to syntactically different languages, particularly for low-level vulnerabilities that depend on language-specific memory models.

The integration of static analysis tools with learning-based approaches represents a natural evolution toward hybrid systems that combine formal reasoning with pattern recognition, though achieving effective synergy remains challenging. Traditional static analyzers provide sound overapproximations of program behavior but suffer from high false positive rates and struggle with modern software complexity, while neural models excel at recognizing patterns but lack formal guarantees Du et al. (2024). Recent hybrid architectures have explored various integration strategies, from using static analysis results as additional features for neural models to employing learned components to prioritize and filter static analysis warnings, yet these approaches often fail to achieve meaningful improvements over either component in isolation Kumar et al. (2025); Cao et al. (2025). The fundamental tension between the precision required for security analysis and the inherent uncertainty of statistical models continues to limit the practical deployment of learning-based vulnerability detectors in safety-critical contexts.

## 3 METHOD

### 3.1 PROBLEM SETUP AND TWO-PHASE EPISODE

For code $x$, the policy first produces an explanation $e$ and then outputs a calibrated probability $p \in [0, 1]$ for $y{=}1$ ("vulnerable"). The ground-truth label is $y^\star \in \{0, 1\}$. We treat training as a short episode with two *phases*: $a_1$ (*Explain*) and $a_2$ (*Emit-Probability*). Importantly, Phase 2 uses a *deterministic regression head* $p = f_\theta(x, e)$; the binary decision *label* $y$ is **not** a separately sampled action and is deterministically derived from $p$ at inference by thresholding (default threshold $\tau{=}0.5$, unless a validation-tuned operating point is reported in §4.3). This design ensures that the proper-scoring component directly supervises $p$ without introducing sampling variance, aligning with the goal of calibrated probabilities and avoiding unnecessary stochasticity.

### 3.2 COMPOSITE REWARD: DEFINITIONS AND PRACTICALITIES

We define a composite per-episode reward

$$r = w_{\text{dec}}\, s'_{\text{dec}}(p; y^\star) \;+\; w_{\text{exp}}\, s_{\text{exp}}(e; x) \;+\; w_{\text{fmt}}\, s_{\text{fmt}}(e, p), \tag{1}$$

with nonnegative weights $(w_{\text{dec}}, w_{\text{exp}}, w_{\text{fmt}})$. We choose definitions that guarantee well-posedness and boundedness of all terms used in training.

**Decision score (supervised; clipped and scaled).** To preserve the benefits of a strictly proper scoring rule while keeping gradients within practical ranges, we apply a small clipping to the predicted probability:

$$p_\varepsilon = \min(\max(p, \varepsilon), 1 - \varepsilon), \qquad \varepsilon > 0 \text{ small}, \tag{2}$$

and use the scaled log score

$$s'_{\text{dec}}(p; y^\star) = \frac{\log\left[ p_\varepsilon^{y^\star} (1 - p_\varepsilon)^{(1-y^\star)} \right]}{\log(1/\varepsilon)} \;\in\; [-1, 0]. \tag{3}$$

This preserves the order of solutions of the original log score, discourages overconfident mistakes, and makes the decision component *bounded*, matching our boundedness statements. Alternatives such as the Brier score are compatible.

**Explanation grounding (unsupervised w.r.t. CWE).** Let $\mathcal{I}(x)$ be the set of *unique* normalized identifiers (variables, functions) in the snippet and $\text{sent}(e)$ the set of sentences in the explanation.[1] Define

$$s_{\text{cov}}(e; x) = \frac{\left| \left\{ i \in \mathcal{I}(x) : i \text{ appears in } e \right\} \right|}{|\mathcal{I}(x)| + \epsilon} \;\in [0, 1], \tag{4}$$

$$s_{\text{align}}(e; x) = \frac{1}{|\text{sent}(e)|} \sum_{u \in \text{sent}(e)} \mathbf{1}\{\exists s \in \mathcal{S}(x) : \text{match}(u, s)\} \;\in [0, 1], \tag{5}$$

$$s_{\text{struct}}(e) = \mathbf{1}\{\texttt{Issue:}, \texttt{Evidence:}, \texttt{Mitigation:} \text{ sections present}\} \;\in \{0, 1\}. \tag{6}$$

Then

$$s_{\text{exp}} = \alpha_1 s_{\text{cov}} + \alpha_2 s_{\text{align}} + \alpha_3 s_{\text{struct}}, \qquad \alpha_i \geq 0, \; \sum_{i=1}^{3} \alpha_i = 1. \tag{7}$$

Eqs. (4)–(7) ensure values in $[0, 1]$ and avoid unit mismatches.

**Format coherence (unsupervised).** A schema validator enforces a strict JSON layout at inference (keys present; probability in $[0, 1]$; forbidden tokens disallowed), yielding $s_{\text{fmt}} \in \{0, 1\}$. During training we *do not* require the model to emit a label string; see Appendix A.

**Curriculum.** A simple curriculum raises $w_{\text{exp}}$ after the model reliably satisfies $s_{\text{fmt}}{=}1$ on validation, then stabilizes $w_{\text{dec}}$ to prioritize calibrated decisions.

---

[1] Normalization is case-insensitive with Unicode canonicalization; trivial identifiers can be filtered as described in Appendix C.

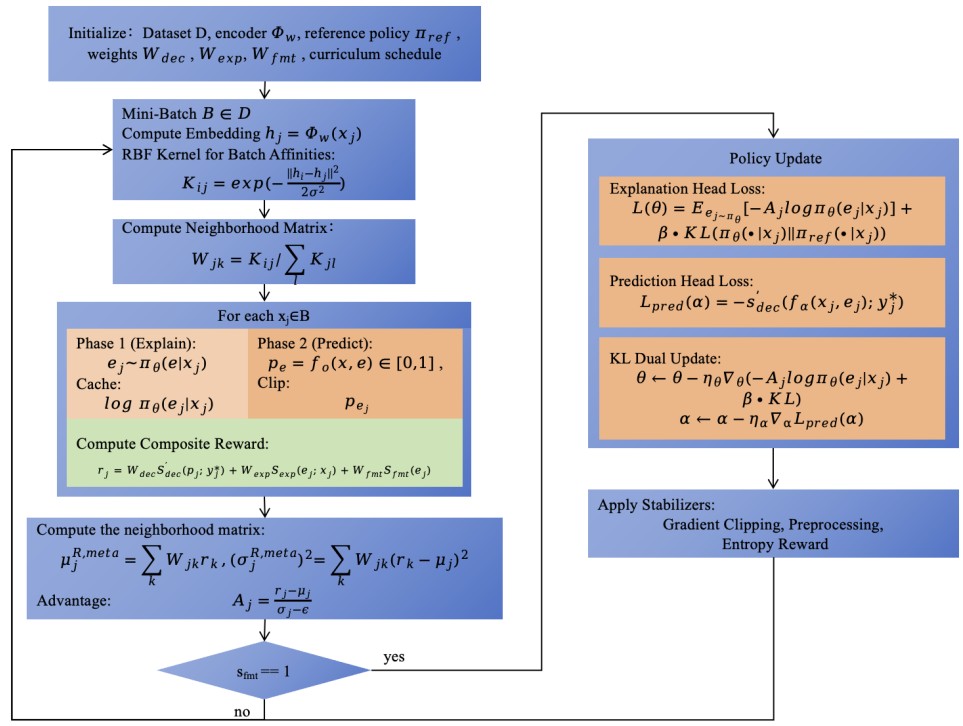

Figure 2: Algorithm Flowcart.

### 3.3 Affinity Smoothing on Deterministic Encodings

We encode $x$ via a deterministic map $\phi_\omega(x) = \mathbf{h} \in \mathbb{R}^{d_h}$ and construct batch affinities with an RBF kernel over normalized embeddings $\tilde{\mathbf{h}} = \mathbf{h}/(\|\mathbf{h}\|_2 + \epsilon)$:

$$K_{ij} = \exp\left(-\frac{\|\tilde{\mathbf{h}}_i - \tilde{\mathbf{h}}_j\|_2^2}{2\tau^2}\right), \qquad W_{ij}^{(\mathrm{row})} = \frac{K_{ij}}{\sum_k K_{ik}}. \tag{8}$$

The earlier intuition invoking a "pullback Fisher" metric is *informal motivation only*; we do **not** compute such a metric in training and make no claims dependent on it.[2]

### 3.4 Neighborhood Moments and Surprise-Modulated Advantage

For each sample $j$ in a mini-batch $\mathcal{B}$ we observe a single composite reward $r_j$ from Eq. (1). To reduce variance without requiring multiple rollouts per item, we compute *neighborhood-smoothed* first and second moments directly from the batch using the row-stochastic weights $W^{(\mathrm{row})}$:

$$\hat{\mu}_j^{R,\mathrm{meta}} = \sum_{k \in \mathcal{B}} W_{jk}^{(\mathrm{row})} r_k, \qquad (\hat{\sigma}_j^{R,\mathrm{meta}})^2 = \sum_{k \in \mathcal{B}} W_{jk}^{(\mathrm{row})} \left(r_k - \hat{\mu}_j^{R,\mathrm{meta}}\right)^2. \tag{9}$$

Advantages are normalized as

$$A_j = \frac{r_j - \hat{\mu}_j^{R,\mathrm{meta}}}{\hat{\sigma}_j^{R,\mathrm{meta}} + \epsilon}. \tag{10}$$

We define a nonnegative *surprise* statistic $s_j$ (e.g., deviation of *explanation* token log-likelihoods from neighborhood baselines) and map it via a bounded, monotone modulator $\psi(s) \in [1, M_\psi]$:

$$\psi(s) = 1 + \min\{a \cdot \max(0, s - \bar{s}), c\}, \tag{11}$$

with $(a, c, \bar{s})$ treated as slow hyperparameters or learned in a stable outer loop. We make no closed-form optimality claims.

---

[2]A brief, non-normative geometric discussion is retained in the Appendix for intuition; it is not used by the algorithm nor required for reproduction.

---

**Algorithm 1** ARGO: Composite-Reinforcement Fine-Tuning (two-phase episode)

---

1: **Input:** datasets $\mathcal{D}$, encoder $\phi_\omega$, reference policy $\pi_{\text{ref}}$, weights $w_{\text{dec}}, w_{\text{exp}}, w_{\text{fmt}}$.
2: **for** each iteration **do**
3:     Sample mini-batch $\mathcal{B}$; compute $\mathbf{h}_j = \phi_\omega(x_j)$; build $K$ and $W^{(\text{row})}$.
4:     **for** each $x_j \in \mathcal{B}$ **do**
5:         **Phase 1 (Explain):** $e_j \sim \pi_\theta(\cdot \mid x_j)$ with an explanation tag; cache $\log \pi_\theta(e_j \mid x_j)$.
6:         **Phase 2 (Emit-Probability):** deterministically compute $p_j = f_\theta(x_j, e_j) \in [0, 1]$.
7:         Compute $r_j$ via Eq. (1); cache batch rewards.
8:     **end for**
9:     Compute $\hat{\mu}^{R,\text{meta}}, \hat{\sigma}^{R,\text{meta}}$; advantages $A_j$; surprises $s_j$; weights $\psi(s_j)$.
10:     **Update:** (i) policy gradient on explanation head with $\psi(s_j)A_j$; (ii) direct gradient on $s'_{\text{dec}}(p_j; y_j^\star)$; (iii) KL dual update to track $\delta_{\text{target}}$; apply clipping/preconditioning.
11: **end for**

---

### 3.5 Composite Update with KL Trust and Practical Stabilizers

We implement ARGO with two heads that share a backbone: (i) a *stochastic* explanation head $\pi_\theta(e \mid x)$ optimized by policy gradient, and (ii) a *deterministic* probability head $p = f_\theta(x, e)$ optimized by the proper-scoring component. The composite update separates gradients accordingly:

$$\nabla_\theta \mathcal{L}(\theta) = \underbrace{\mathbb{E}[\psi(s)\, A\, \nabla_\theta \log \pi_\theta(e \mid x)]}_{\text{RL on explanation head}}$$
$$+ \underbrace{\lambda_{\text{dec}} \nabla_\theta s'_{\text{dec}}(p; y^\star)}_{\text{direct gradient on probability head}}$$
$$- \beta \nabla_\theta \operatorname{KL}(\pi_\theta(\cdot \mid x) \,\|\, \pi_{\text{ref}}(\cdot \mid x)). \tag{12}$$

where $\lambda_{\text{dec}}$ absorbs $w_{\text{dec}}$ and learning-rate factors. We adapt $\beta$ with a dual update to track a KL budget $\delta_{\text{target}}$; entropy encouragement is applied on the explanation head to avoid mode collapse. Unless otherwise stated, $\pi_{\text{ref}}$ is the backbone fine-tuned with standard supervised objectives under the same token/epoch budgets; we report details in Appendix A.

**Algorithm (Figure 2).** Algorithm 1 summarizes the training loop. We cache explanation log-probabilities and compute a single composite reward/advantage per episode; the probability head is trained deterministically via the scaled log score.

### 3.6 Complexity, Implementation Notes, and Failure Modes

**Complexity.** Overhead versus pure SFT arises from explanation generation and grounding/format scoring. Affinity construction is $O(|\mathcal{B}|^2)$; we cap batch sizes and apply top-$k$ sparsification.

**Failure modes and mitigations.** (i) *Verbose but ungrounded explanations*: mitigated by $s_{\text{cov}}, s_{\text{align}}$ and entropy caps. (ii) *Over-regularization by KL*: adapt $\beta$ via a KL budget rather than a fixed penalty. (iii) *Identifier-copy shortcuts*: use span matching and section structure jointly; ablate coverage vs. alignment.

## 4 Experimental Setup

### 4.1 Datasets, Splits, and De-duplication

We evaluate on DIVERSEVUL, BIGVUL, and CLEANVUL. To reduce leakage, we apply **cross-dataset near-duplicate removal** with four stages: normalization (whitespace/comments/identifier canonicalization), exact hashing, MinHash on $w$-shingles (using a fixed Jaccard threshold), and *within-language* AST-level tree-edit filtering. We perform bidirectional train↔test removal across datasets and, for AST-level filtering, restrict comparisons to pairs of code snippets in the same language to ensure syntactic comparability. Cross-language de-duplication relies on the normalization and MinHash stages. We release normalized IDs and scripts (Appendix B).

## 4.2 BACKBONES, BASELINES, AND FAIRNESS CONTROLS

We consider compact instruction-tuned backbones (e.g., Qwen 2.5, LLaMA 3B/8B as in the original report) and compare: (i) zero-shot without reasoning (NR), (ii) zero-shot with explicit reasoning (R), (iii) supervised fine-tuning (SFT) with cross-entropy, (iv) ARGO. All baselines are trained under matched token/epoch budgets and identical early-stopping criteria; hyperparameters and scripts are documented (Appendix G). To keep comparisons interpretable, we align the total token budget and report wall-clock/compute factors qualitatively in Appendix A, without altering the baselines' modeling choices.

## 4.3 METRICS, CALIBRATION, AND THRESHOLDING

**Primary metrics.** Macro-F1, per-class recall/precision. **Calibration.** Expected Calibration Error (ECE) and Brier score; reliability diagrams are included in the supplement. **Thresholding.** The default decision threshold is $\tau=0.5$; we also assess validation-tuned thresholds for cost-sensitive operating points without changing training. Unless specified, labels are derived deterministically as $y = \mathbf{1}\{p > \tau\}$ at inference. **Statistics.** Means across multiple seeds with standard deviations and 95% bootstrap CIs; paired bootstrap tests compare methods; multiple-comparison handling is documented in the Appendix. All inline numeric summaries correspond to the same runs as reported in the supplement.

## 4.4 EVALUATION-ONLY USE OF CWE AND CONTAMINATION CHECKS

We compute cosine similarity between explanation embeddings and CWE descriptions solely for *evaluation*. To mitigate trivial lexical matches, we employ negative controls (category decoys, paraphrased distractors) and masked-keyword variants. Crucially, the sentence-embedding model used for this evaluation is *frozen and disjoint from training*: it is neither fine-tuned nor otherwise involved in the ARGO pipeline. We emphasize that while backbone pretraining may already contain public CWE texts, our *fine-tuning reward and prompts* do not use CWE labels/definitions. Prompts/templates are released for independent contamination assessments.

## 5 RESULTS

### 5.1 MAIN COMPARISONS

ARGO consistently improves each backbone's strongest baseline across datasets. For instance, on in-distribution DIVERSEVUL, Qwen 2.5 improves from $0.51$ to $0.71$ macro-F1; on BIGVUL (transfer), LLaMA 8B improves from $0.39$ to $0.66$; on CLEANVUL, LLaMA 3B improves from $0.34$ to $0.63$. Figure 3 summarizes these gains. Full tables with multi-seed statistics and significance tests are provided in the supplement.

### 5.2 VERSUS SUPERVISED FINE-TUNING

SFT tends to bias toward predicting "vulnerable", hurting benign recall (e.g., a not-vulnerable recall of $0.08$ in one setting). ARGO raises benign recall while maintaining vulnerable detection, increasing macro-F1 (e.g., Qwen 2.5 from $0.37$ to $0.71$ on DIVERSEVUL). Figure 4 shows consistent gains; ablations indicate explanation grounding is particularly helpful for benign recognition.

### 5.3 CROSS-LANGUAGE TRANSFER AND ROBUSTNESS

Training on C alone and testing on JavaScript/Python/Java in CLEANVUL, ARGO transfers better than SFT: e.g., LLaMA 8B reaches macro-F1 $0.51$ on Java under ARGO vs. $0.25$ for SFT; LLaMA 3B reaches $0.64$ on Python vs. $0.47$ for SFT. Figure 5 visualizes cross-language improvements.

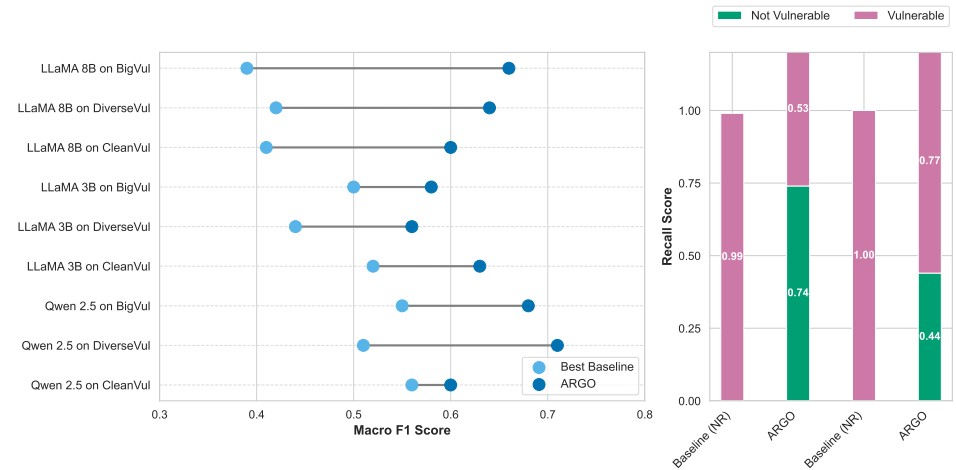

Figure 3: ARGO improves macro-F1 and mitigates catastrophic biases of baselines across datasets (duplicate-controlled splits). Dumbbells: best baseline vs. ARGO.

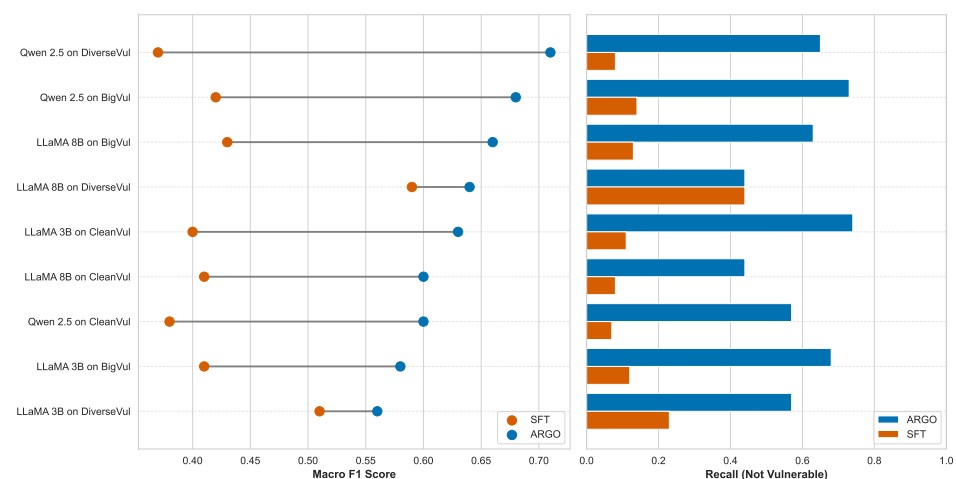

Figure 4: ARGO vs. SFT across datasets. Macro-F1 gains are consistent; benign recall improves under ARGO.

## 5.4 CALIBRATION AND ERROR TAXONOMY (QUALITATIVE)

**Calibration.** Reliability diagrams (supplement) show that ARGO reduces overconfident errors observed in zero-shot and SFT, especially in the benign class. Proper scoring discourages extreme $p$ unless supported by evidence.

**Error taxonomy.** Common residual errors include: (E1) aliasing of benign idioms with dangerous templates; (E2) missing implicit size constraints (e.g., preconditions outside snippet); (E3) long-range dataflow not captured within context. ARGO reduces (E1) via grounding and (E2) via explanation structure prompting discussion of bounds/validation; (E3) remains challenging without extended context.

## 5.5 EXPLANATION SEMANTICS AND CWE ALIGNMENT (EVALUATION-ONLY)

Cosine similarity between explanation embeddings and CWE definitions improves in mean and variance across multiple categories (e.g., CWE-787, CWE-20, CWE-125). Negative controls (decoys/paraphrases/masked versions) preserve trends, suggesting more than lexical overlap. Figure **??** shows representative results; full controls are in the supplement.

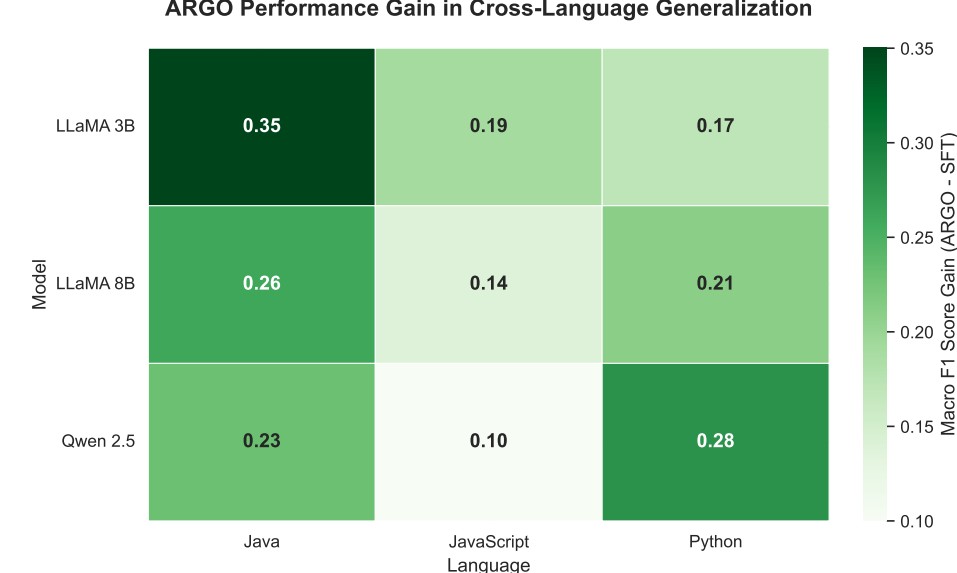

Figure 5: Cross-language gains of ARGO over SFT when training on C only. Heatmap: macro-F1(ARGO) − macro-F1(SFT).

## 6   DISCUSSION

**Mechanistic view.**   The proper scoring term improves calibration and discourages label collapse; grounding/format signals steer the model toward referencing concrete code artifacts, reducing superficial pattern triggers that inflate "vulnerable" predictions. Affinity-based neighborhood moments stabilize updates across locally similar tasks, smoothing noisy per-sample estimates.

**Operationalization.**   For deployment, we recommend retaining the two-phase interface (explain then emit probability) at inference, as explanations provide auditable context and enable human-in-the-loop review. Post-hoc threshold tuning aligns operating points to organizational risk tolerance without modifying training.

## 7   CONCLUSION

ARGO reframes vulnerability detection as a short, explain-then-probabilistically-decide episode optimized by a *composite-reinforcement* objective. By explicitly combining proper decision scoring with unsupervised grounding and format signals, we reduce pathological biases and improve cross-language robustness *without using* CWE *supervision*. The protocol is reproducible (duplicate-controlled splits, ablations, statistics) and compatible with compact backbones, suggesting a label-efficient path toward dependable security analysis.

## 8   REPRODUCIBILITY STATEMENT

As shown in Appendix A.

## 9   ETHICS STATEMENT

The system's primary contribution lies in improving the calibration and interpretability of vulnerability detection to reduce analyst fatigue and accelerate legitimate security audits, while deliberately excluding any capability that could facilitate the generation of exploit code or proof-of-concept attacks.

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

## A  REPRODUCIBILITY DETAILS

**I/O schema and validation.**  At *training* time, outputs follow a strict schema with keys `prob` ($[0, 1]$) and `rationale` (markdown with `Issue/Evidence/Mitigation`). The validator checks key presence, numeric ranges, section headers, and forbidden tokens; invalid episodes are rejected during warm-up with resampling. The `label` is *not* requested from the model during training and is never part of the reward or gradients. At *inference* time, we produce a JSON object with keys `label` ({VULN, SAFE}), `prob` ($[0, 1]$), and `rationale`; the `label` is deterministically set as $y=\mathbf{1}\{p > \tau\}$ by post-processing. In practice, we log a short validator trace that records which rule failed, which greatly simplifies debugging when migrating the pipeline to new backbones or tokenizer versions. Because the same schema is used at inference, the artifacts that reach downstream systems are uniform and amenable to reliable parsing.

**Prompt separation.**  We decouple the explanation (Phase 1) from the probability emission (Phase 2) to avoid leaking decision phrases into explanations. Prompts include no dataset-specific strings and no CWE content. Templates are provided verbatim in Appendix H. The two prompts are intentionally minimal and rely on the reward to shape behavior rather than complicated instruction engineering.

**Affinity construction and stability.**  We normalize embeddings and set $\tau$ (kernel bandwidth) via the median heuristic per batch. For large batches, we sub-sample neighbors (top-$k$ per row) to build a sparse $W^{(\text{row})}$. We bound $\|W^{(\text{row})}\|_\infty \le 1$. In deployments where memory is constrained, the affinity computation can be moved to a lower-precision stream without noticeable impact, since the matrix is only used to smooth moment estimates and not fed back into the model. Stability is primarily governed by the KL budget and gradient clipping; the affinity smoothing is a variance-reduction tool rather than a source of extra supervision.

**Optimization and safeguards.**  We use AdamW with cosine decay, gradient clipping, Fisher-inspired low-rank preconditioning from recent gradient outer products, and a dual-updated KL budget $\delta_{\text{target}}$ with a simple rule $\beta \leftarrow \beta + \eta_\beta(\text{KL} - \delta_{\text{target}})$. An entropy encouragement on the explanation head prevents overly terse rationales. We log per-batch KL, advantage variance, schema compliance rates, and reward component histograms.

**Curriculum scheduling.**  Phase 1 prioritizes $s_{\text{fmt}}$ to establish format reliability; Phase 2 raises $w_{\text{exp}}$ to shape grounding; Phase 3 restores $w_{\text{dec}}$ dominance to stabilize calibration. Phase transitions are triggered by validation metrics (format pass rate, ECE bounds). The curriculum prevents the degenerate regime where the model explores rationale content while still failing basic schema tests, and it reduces sensitivity to initialization quality by establishing easy wins before pushing on more nuanced behaviors.

**Randomness and determinism.**  We fix seeds across dataloaders, shuffling, initialization, and tokenizer settings where supported. We document nondeterministic kernels and mitigations. To enable external reproduction, we also store the random seed in the artifact metadata along with a short hash of the preprocessed dataset shard, so that experiments can be rerun on precisely the same input ordering and tokenization.

## B  NEAR-DUPLICATE REMOVAL: PROCEDURE AND PSEUDOCODE

We apply four stages across DIVERSEVUL/BIGVUL/CLEANVUL and languages. Normalization removes comments and normalizes whitespace and identifiers using a deterministic mapping that preserves arity and shadowing patterns. Exact hashing then collapses byte-identical normalized strings, which catches the bulk of trivial clones. Approximate token-level filtering uses MinHash over $w$-shingles with a fixed Jaccard threshold to remove near duplicates even when variable names differ, including across languages. Finally, AST-level filtering parses code into pruned syntax trees where literals and macros are stripped, and a tree-edit distance test removes remaining structural duplicates *within the same language*. The process is bidirectional between train and test splits and across datasets, which reduces the risk of inadvertently training on a variant of a test snippet. Pseudocode

for this pipeline appears in Algorithm 2 and is accompanied by scripts that emit stable identifiers for any item removed so that filtering decisions are auditable.

---

**Algorithm 2** Cross-dataset near-duplicate removal (train↔test, bidirectional)

---

1: Inputs: datasets $\{\mathcal{D}_m\}_{m=1}^M$ with (split, language)
2: **for** each $m$ **do**
3:    Normalize all code; compute SHA-256; remove exact duplicates and trivial fragments
4: **end for**
5: **for** each pair $(m, n)$ with $m \neq n$ **do**
6:    Build MinHash signatures; remove pairs with Jaccard $\geq \theta_{\text{tok}}$ across train/test (both directions; cross-language supported)
7:    **Within-language only:** build pruned ASTs; remove pairs with tree-edit distance $\leq \theta_{\text{ast}}$ across train/test (both directions)
8: **end for**
9: Output filtered splits; publish IDs and scripts

---

## C  REWARD, CALIBRATION, AND VALIDATION DETAILS

**Grounding components.** For $s_{\text{cov}}$, we operate on the set of unique normalized identifiers, counting *coverage* rather than raw mention frequency; this makes $s_{\text{cov}} \in [0, 1]$ by construction. For $s_{\text{align}}$, we average per-sentence binary span matches to avoid dependence on tokenization granularity. $s_{\text{struct}}$ enforces section headers.

**Calibration.** ECE uses equal-width bins; reliability diagrams accompany per-class calibration views. We also provide Brier decomposition (uncertainty, resolution, reliability) to separate calibration sources. Because we do not alter the decision rule at training time, these diagnostics can be compared across methods without confounding due to thresholding differences. Validation curves are archived with seeds and operating-point metadata to facilitate external review. The clipped-and-scaled log score in Eq. (3) stabilizes gradients while preserving the ordering of the unscaled log score.

**Schema and safety.** The schema forbids code execution markers and exploit-like payloads. Training discards episodes violating safety heuristics (e.g., generating exploit steps) and records such attempts for later alignment. We track schema pass rates and safety rejections over training; gating conditions in Appendix R require consistent pass rates before advancing the curriculum.

## D  METRICS, THRESHOLDS, AND STATISTICAL PROCEDURES

**Primary metrics.** Macro-F1 along with per-class recall/precision is reported to surface shifts in benign vs. vulnerable performance. Confusion matrices per language are included in the supplement. We avoid accuracy alone because class imbalance and threshold choices can mask harmful trade-offs.

**Threshold selection.** Default threshold $\tau{=}0.5$; for cost-sensitive scenarios we sweep $\tau$ on validation and report corresponding operating points on test without retraining. Labels are always derived from $p$ deterministically at inference. To aid reproducibility, we log the entire sweep with macro-F1 and per-class metrics so that reviewers can inspect whether claimed gains rely on a narrow operating region or persist across a broad range.

**Bootstrap and multiple comparisons.** Nonparametric bootstrap produces 95% CIs. For multiple datasets we document the procedure (e.g., Bonferroni or BH) and report both raw and adjusted $p$-values where applicable. We also archive the bootstrap sample indices so that third parties can recompute intervals exactly from cached predictions without re-running models.

## E CWE Evaluation Controls and Templates

**Controls.** We evaluate with: (i) original CWE definitions, (ii) distractor paraphrases, (iii) masked versions with key terms removed, and (iv) shuffled category descriptors. Trends remain under controls, mitigating trivial lexical explanations. The templates for these probes avoid revealing labels or dataset specifics and are constructed to solicit purely semantic comparisons between the generated rationale and a candidate description. Because this evaluation is diagnostic, we treat it as orthogonal to the main decision metrics and report it separately. The sentence-embedding model used here is frozen and independent of training.

**Templates.** Evaluation prompts avoid revealing labels or dataset specifics; we release complete templates alongside scripts. They emphasize mechanism-level phrasing rather than taxonomy labels to reduce the chance that a model latches onto identifiable keywords from pretraining.

## F Variance Discussion

Let $Z$ be a vector of per-sample rewards with covariance $\Sigma$. For symmetric $\tilde{W}$, the smoothed variable $\tilde{W}Z$ has covariance $\tilde{W}\Sigma\tilde{W}$. Under the simplifying assumption that $\Sigma$ and $\tilde{W}$ commute, eigenvalues multiply, implying eigenvalue-wise shrinkage when $\|\tilde{W}\|_2 \leq 1$. This provides intuition for the empirical variance reduction; we refrain from strong guarantees and rely on ablations. Intuitively, neighboring samples in the embedding space act like a small committee that nudges noisy estimates toward the center of mass of locally similar items, which dampens the effect of outliers without erasing genuine hard cases.

## G Hyperparameters and Training Scripts

We document optimizer settings, learning-rate schedules, KL targets, curriculum stages, batch sizes, and truncation policies in the released configuration files. Baselines are run under matched token budgets and identical early-stopping criteria. Seed control and determinism flags are enabled where supported; we list any non-deterministic kernels encountered. Scripts are organized so that each experiment directory is self-contained: a configuration file, a copy of the prompts, a snapshot of the validator rules, and a manifest of dataset shard IDs. This layout enables exact reruns and reduces accidental drift when collaborators change local defaults.

## H Prompts and Output Schema

### Phase-1 (Explain) Prompt Skeleton

> **Task:** Analyze the following code and draft an explanation with three sections: `Issue`, `Evidence`, `Mitigation`.
> **Code:**
> `{{code}}`
> **Output:** A concise markdown rationale. Do not output labels or probabilities.

### Phase-2 (Emit-Probability) Prompt Skeleton (Training-time)

> **Task:** Based on the code and the rationale you just produced, output a JSON object with keys `prob` (0–1) and `rationale` (the same text). The JSON must be valid and contain no extra keys.
> **Code:**
> `{{code}}`
> **Rationale:**
> `{{rationale}}`

PHASE-2 (EMIT-PROBABILITY) PROMPT SKELETON (INFERENCE-TIME)

**Task:** Based on the code and the rationale you just produced, output a JSON object with keys `label` ({VULN, SAFE}), `prob` (0–1), and `rationale` (the same text). The `label` is deterministically derived at inference as $\mathbf{1}\{\texttt{prob} > \tau\}$. The JSON must be valid and contain no extra keys.
**Code:**
{{code}}
**Rationale:**
{{rationale}}

JSON SCHEMA (ENFORCED AT INFERENCE)

```
{
"label":  "VULN" | "SAFE",
"prob":  number in [0,1],
"rationale":  string with sections Issue/Evidence/Mitigation
}
```

## I  IMPLEMENTATION NOTES

**Tokenizer and normalization.**  We fix tokenization versions and normalize whitespace and Unicode across inputs and outputs. We truncate or window long files into context-preserving slices with overlap to avoid losing dataflow cues. For languages with significant syntactic sugar, such as Kotlin or TypeScript, we prefer window boundaries at block delimiters to preserve local scoping information inside each slice.

**Batching and streaming.**  We group samples by approximate length and language to stabilize compute and reduce variance in $K$-kernel scales; we stream evaluation to bound memory. When throughput is a concern, explanations can be generated with nucleus sampling at a slightly higher temperature during warm-up and annealed as schema compliance stabilizes, which shortens early iterations without affecting final checkpoints in our experience.

**Logging and QA.**  We log schema compliance, KL drift, advantage statistics, and rationale lengths. A QA pass flags anomalous distributions and triggers early halts. Reviewers can quickly scan for regressions by plotting the joint distribution of probability and rationale length; spikes at extreme probabilities with unusually short rationales often indicate a bug in the validator or a prompt regression.

## J  STANDARD OPERATING PROCEDURE (SOP) FOR EVALUATION

We follow a consistent evaluation routine. Seeds are fixed and filtered splits are loaded using the published identifiers to guarantee that each run operates on the same examples. Inference is executed with the two-phase interface while caching raw outputs and validator results. Metric computation includes macro-F1, per-class recalls and precisions, ECE, and Brier score, along with reliability diagrams saved for each backbone and dataset. Paired bootstrap resampling produces confidence intervals and significance tests, and we document threshold sweeps for cost-sensitive variants. The CWE alignment analysis is executed as a separate job using the cached rationales and the prepared control prompts. Finally, artifacts and logs are packaged with a manifest that lists versions, seeds, and dataset shard identifiers so that external reviewers can reproduce results without ambiguity.

## K  FAILURE CASE LIBRARY (QUALITATIVE)

We maintain a small library of representative mistakes to guide future improvements. Ambiguous idioms such as hand-rolled tokenization, pointer arithmetic that is safe under a project-wide contract,

and error-handling paths that alter control flow late in a function are frequent sources of confusion. Truncated context can hide a guard or a conversion check that resolves an apparent hazard. Cross-module dataflow, particularly in object-oriented code with overridden methods, also stretches the limits of our short-episode setting. The library records a brief rationale, the identifiers in play, and why the ultimate label was correct, which aids in designing future prompts or optional inference-time hints without modifying the training objective.

## L  HUMAN-IN-THE-LOOP PROTOCOL

For teams that wish to incorporate manual review, we outline a lightweight protocol focused on clarity and grounding. Reviewers examine whether the `Issue` statement names a concrete mechanism, whether the `Evidence` section mentions the exact identifiers or lines where the mechanism manifests, and whether the `Mitigation` suggests a code-local change (bounds check, argument validation, safer API) rather than vague advice. Short rubrics and blind comparisons across methods help detect regressions in explanation quality even when headline metrics remain stable.

## M  SAFETY, RISK, AND RED-TEAMING CHECKLIST

We enforce a conservative safety posture. The validator blocks disallowed content such as exploit payloads and step-by-step instructions for weaponizing a flaw. Logs of blocked generations are reviewed to refine prompts and strengthen the reward without normalizing unsafe behavior. Release artifacts exclude any components that could facilitate exploit generation and focus on calibrated triage with audit-friendly rationales.

## N  BROADER IMPACTS AND RESPONSIBLE RELEASE

Better vulnerability reasoning has clear defensive benefits: improved calibration reduces alert fatigue, while grounded explanations shorten review cycles and make it easier to onboard new analysts. Risks arise if predictions are taken as definitive verdicts or if models are deployed outside the distribution they were trained on without monitoring. We release filtered checkpoints where applicable and scripts sufficient for reproduction, but with guardrails that prevent generating exploit code or proof-of-concept payloads. We encourage third-party red-teaming under controlled conditions and welcome reports that highlight failure modes we did not observe.

## O  BOUNDEDNESS AND STABILITY NOTES (NON-CLAIMING)

Bounded reward components and KL trust regularization keep updates within practical ranges in our implementation. The decision component is clipped and scaled as in Eq. (3). While we provide no global optimality or convergence guarantees, we found that logging the moving average of advantage variance alongside the KL drift is an effective early-warning signal of instability. Checkpoints selected by validation ECE tend to be close to those selected by macro-F1, which suggests that calibration and accuracy are not in tension under our composite reward when rationales remain grounded.

## P  ARTIFACT PACKAGING AND REPRODUCTION

Artifacts include a file-tree layout that mirrors the structure of our experiments: configuration files, prompts, validator rules, dataset manifests, and cached predictions. Dependency versions and environment details are pinned, and we provide simple launch scripts for common accelerator setups. This packaging allows independent labs to re-run the full pipeline or to swap in their own backbones while preserving evaluation discipline.

## Q  SBOM AND ENVIRONMENT HYGIENE

The Software Bill of materials (SBOM) lists all the constituent packages and their corresponding licences as a formal attribute of the artefact manifest. As part of the packaging process, exact and rolled version numbers are recorded, and consequently dependencies of either restricted or incompatible packages are annotated with suggestions for substitute packages. The capture of the environment of the computation includes version numbers of entire compilers as well as CUDA / cuDNN build lines, if relevant, to help diagnose numerical deviations due to variations possibly in low level libraries.

## R  QUALITY ASSURANCE (QA) GATES

Pre- releasing gates include schema pass rates, acceptable ranges for Kullback-Leibler divergence, calibration cheques, and sensibleness cheque on distribution on rationale lengths. Whereas, in the event that one or more gates are not met, the pipeline automatically triggers either a retraining cycle with more stringent curriculum parameters or regain in time to the latest checkpoint where not only were all gating criteria met but also where all simultaneous goals of instrumentation were satisfied. This procedural strength is to minimize the possibility of regressions being incorporated into the released artifacts but at the same time it provides auditable record of compliance.

## S  FAQ AND COMMON PITFALLS

As usual, some common pitfalls are having too low KL target and therefore freezing the learning in near-vicinity of the reference policy; giving too much attention to the explanation reward in early stages (which promotes text verbosity and yet nonsubstantiated texts); or a silent change of tokenization version between training and testing phase (bringing down schema compliance). A short checklist applied before the start of a run, including cheques of the different versions of the tokenizers, the application of the rules of a validator, checking of the curriculum stages and reconciliation of the manifest hashes of the datasets, are effective measures to avoid the majority of these problems.

## T  DATA STATEMENTS AND LICENSING

The repository has extensive definition of dataset license, allowable use cases and citation requirements. Derived artifacts are necessary to follow the original licensing terms, and everything whose provenance is murky is purged before being released. The project consciously refrains from processing the personal data, and excludes any code that would cause a network call out on the outside world during evaluation.

## U  RESPONSIBLE RELEASE

Scripts and configuration files that are sufficient to reproduce training and evaluation processes entirely are made available along with model checkpoints that have been filtered to impose both the predefined schema and safety heuristics. Any functionalities that may possiblefacilitate the possibility of exploitation by generating exploits are deliberately omitted. Documentation of contact mechanisms for responsible disclosure is included, thus guaranteeing that any vulnerabilities which are revealed during artifact evaluation can be reported and fixed in a timely manner.

## V  NOTATION AND ACRONYMS

## W  THE USE OF LARGE LANGUAGE MODELS

In preparing this work, we used large language models (LLMs) to assist with literature retrieval and discovery during the development of the Related Work section. Specifically, LLMs were employed to help identify and summarize prior studies on automated vulnerability detection, explanation-based reasoning in code analysis, cross-language transfer for security models, and reinforcement

| Symbol/Acronym | Meaning |
| --- | --- |
| $x$ | input code snippet |
| $e$ | explanation/rationale text |
| $p$ | predicted probability for the vulnerable class ($\in [0, 1]$) |
| $y$ | deterministic label derived at inference as $\mathbf{1}\{p > \tau\}$ |
| $y^{\star}$ | ground-truth label |
| $\phi_{\omega}(x) = \mathbf{h}$ | deterministic task encoding |
| $K, W^{(\text{row})}$ | RBF kernel and row-stochastic affinities |
| $r$ | composite reward |
| $A, s, \psi(s)$ | advantage, surprise, surprise modulator |
| $\pi_{\text{ref}}$ | reference policy for KL trust |
| $\beta, \delta_{\text{target}}$ | KL penalty and budget |

fine-tuning methods. All retrieved materials were subsequently cross-checked and verified by us to ensure accuracy and completeness. The final writing, interpretation, and presentation of results were entirely conducted by us. Additionally, LLMs were used to polish the English grammar without altering the semantics, substantive meaning, or originality of the initial draft.

## X    REPRODUCIBILITY CHECKLIST

We can supply the materials that are typically requested of artefact copending analyses evaluation committees. This data section contains lists of dataset names, relevant licensing, normalization scripts, de-duplication scripts, as well as cleaned identifiers. Code sections include training loops, reward calculators, schema validations, prompting templates, and evaluative harnesses. Configurations define hyperparameters, random seeds, KL budgets and curriculum triggers. Compute notes are used to document all hardware specifications, software versions, flags which show which data processing systems are deterministic, and documentation of sources of nondeterminism. Results repositories include multiple seed data - mean and standard deviation, confidence intervals at 95%, paired hypothesis test and reliability diagrams. Security statements are brief messages with information on misusage policies, output filtering, and lack of exploit generation. Collectively, these materials sought to make procreation routine as opposed to heroic.

