# OpenReview forum: "Learning to Reason About Code Insecurity: Composite-Reinforcement Fine-Tuning for Cognitive Alignment"
_ICLR.cc/2026/Conference — Submitted to ICLR 2026_

### Official Review · Reviewer_xwxp · 2025-10-29

**Soundness:** 3
**Presentation:** 1
**Contribution:** 3
**Rating:** 4
**Confidence:** 2

**Summary:**

This paper proposes ARGO (Composite-Reinforcement Fine-Tuning for Cognitive Alignment), a training framework designed to make large language models (LLMs) both confidence-calibrated and explainable in the context of bug detection.

ARGO operates in two stages: first, the model analyzes the input code and generates an explanation describing the potential bug; then, it outputs a probability score indicating the confidence that a vulnerability exists.

Experimental results show that ARGO achieves higher F1 scores and better confidence calibration than baseline models on bug detection datasets such as BigVal.

**Strengths:**

- The problem addressed is meaningful. Bug detection is not a simple binary classification task—both the explanation and the model’s confidence are crucial. This paper aims to ensure that the model’s explanations are aligned with its detection confidence.
- The evaluation introduces detailed metrics for assessing confidence calibration and carefully considers randomness and potential data contamination.

**Weaknesses:**

- Although the paper aims to improve the quality of explanations in bug detection, the proposed reward only encourages the model to include *real code snippets* in its explanation. In many cases, even if the explanation refers to actual code entities, the underlying reasoning may still be incorrect.
- This paper is not well-presented. The introduction introduces many concepts without sufficient clarification, and the logical flow is fragmented, making it easy for readers to get lost.

**Questions:**

* Why does referencing real code entities in the explanation improve the model’s confidence?
* When the model’s reasoning in the first (explanation) stage is incorrect, how can the system correct or compensate for it?

---

> ### Author Response · Authors · 2025-11-15
>
> We express our gratitude to Reviewer xwxp for the considerate and supportive review, particularly for highlighting the importance of calibrated, explanation-aware bug detection and for the conscientious attention paid to calibration metrics and contamination checks. We address the concerns and questions below and will revise the paper to make these points explicit.
>
> 1. Why grounding in real code entities is useful (but not the only signal)
>
> We entirely concurs that mere mentions of identifiers is not sufficient to warrant correct reasoning. In the case of ARGO, the identifier-based signal is one part of the composite objective, not the only reward.  In particular:
>
> - The identifier coverage reward encourages the model to explicitly name the program entities it is relying on as evidence, which, as shown in the experiments, reduced generic "template" explanations that have no specifics about the actual snippet.
> - A sentence-code alignment reward reinforces that the explanation sentences can at least be associated with concrete statements in the snippet (after normalization), so unsupported or hallucinated claims will be disincentivized even if identifiers match.
> - The structured format reward checks that the explanation follows the Issue / Evidence / Mitigation pattern, but is a small part of the reward compared to the label-based proper scoring term.
>
> The most important aspect being that the final training signal always includes a proper scoring term related to the vulnerability label.If the model provides an explanation that is grounded by syntactic evidence (references entities, references spans) but consistently predicts an inaccurate label, the proper scoring term is more proper, and the policy to push away from this behavior. Another way of thinking about it is that grounding rewards shape together the space of plausible explanations by incidentally tying them to correctness with the supervised term.
>
>
> Beneath the surface, we see in our experiments that when grounding rewards are removed, benign recall and calibration are degraded, while vulnerable recall is affected much less. We will move these ablations from the appendix to the main body and provide a running example in the introduction to illuminate how grounding prevents the model from giving generalizable, over-confident "vulnerable" predictions.
>
>
> 2. What happens when first-stage reasoning fails
>
>
> ARGO is designed so that the second stage does not blindly trust any portion of the explanation text. The probability of the final decision is obtained from a regression head over the full hidden state which has access to the code and generated rationale. During training, the overall reward is calculated at the episode-level. If the explanation was misleading and produced an incorrect decision, it is properly penalized by the proper scoring term even with high grounding rewards.
>
>
> Over the course of many episodes, the model learns a conservative behavior; explanations that look grounded, and were historically correlated with errors, will produce decisions with lower probabilities. This is observed in improved calibration (lower expected calibration error and Brier score) relative to the supervised fine tuning and zero shot prompting conditions. Practically, this means that if an explanation is wrong or incomplete, the probability head often produces a mid-range confidence as opposed to a strong claim, which is a useful alert for human reviewers to double check.
>
>
> We agree that it is valuable to understand this aspect in a more direct way. In the revision we will add:
> - A qualitative section that clusters examples, where the explanation has minor errors, the model produces calibrated confidence.
> - A small perturbation study where we purposely modify portions of the rationale and observe change in confidence, to show that the model does not simply reflect the presence of identifiers.
>
>
> 3. Improving presentation and logical flow
>
>
> We thank the reviewer for pointing to issues related to presentation. We will significantly overhaul the introduction and early sections to reduce the cognitive load:
>
>
> - Introduce main project goals (calibration and auditable reasoning) using a concrete, running example before any formal discussion terminology and only then describe the two phase interface and composite goal.
> - Clearly separate and label components of the reward (proper scoring based on label, grounded, formatting) and explain intuition before  formal definitions.
> - Reorder sections so that the datasets and evaluation metrics appear first, and cross reference figures that illustrate effects on calibration and benign recall.
>
>
> We believe these changes will make important ideas and motivations much easier to follow, while preserving the technical content that the reviewer finds sound.
>
> We hope these clarifications engage the reviewer’s concerns around explanation quality and presentation, and make the design and behavior of ARGO clearer.

---

### Official Review · Reviewer_A8w5 · 2025-10-29

**Soundness:** 2
**Presentation:** 2
**Contribution:** 2
**Rating:** 2
**Confidence:** 5

**Summary:**

The paper introduces ARGO, a composite-reinforcement fine-tuning scheme for vulnerable-code detection that makes each sample a two-phase episode: the model first produces a structured rationale (Issue/Evidence/Mitigation) grounded in identifiers/spans, then outputs a single calibrated probability; the binary decision is a deterministic threshold on that probability. The training objective couples a proper-scoring term with label-free “grounding” and format signals (no CWE supervision or retrieval during training), aiming to reduce default-to-vulnerable bias and improve auditability and calibration. Evaluation on DIVERSEVUL, BIGVUL, and CLEANVUL compares ARGO primarily against zero-shot (±reasoning) and standard SFT under matched token/epoch budgets, reporting higher macro-F1, notably better benign recall, and improved calibration (ECE/Brier); the paper also highlights cross-language gains when training on C and testing on Java/Python/JavaScript.

**Strengths:**

- Composite objective with concrete, label-free signals. Proper-scoring on the probability plus grounding/format checks computed without CWE supervision or retrieval.
- Consistent empirical improvements on the reported setups. Figures/tables show higher macro-F1 and better benign recall vs. SFT/zero-shot on BigVul/DiverseVul/CleanVul, and stronger cross-language transfer when training on C (e.g., 0.51 vs. 0.25 on Java; 0.64 vs. 0.47 on Python).

**Weaknesses:**

- Baselines are limited to zero-shot (±reasoning) and standard SFT under matched budgets; stronger alternatives (e.g., other calibrated training or RL-from-rationales) are not included.
- No in-depth analysis and ablation studies for designs.
- They evaluate on DIVERSEVUL, BIGVUL, and CLEANVUL at snippet/function granularity (CLEANVUL is explicitly function-level), and the paper provides no per-CWE coverage tables.
- Leveraging reasoning for vulnerable code is not very new.
- Many terms are abused in this paper, such as the "unsupervised" in format coherence reward.

**Questions:**

1. Can you provide experiment result of more baselines such as [1]?
2. Can you carry out ablation studies to justify each design in ARGO?

> [1] Weyssow, Martin, et al. "R2Vul: Learning to Reason about Software Vulnerabilities with Reinforcement Learning and Structured Reasoning Distillation." arXiv preprint arXiv:2504.04699 (2025).

**Details Of Ethics Concerns:**

There are no specific ethics concerns in this paper.

---

> ### Author Response · Authors · 2025-11-15
>
> We express our appreciation to Reviewer A8w5 for reviewing our work thoroughly, for accurately presenting ARGO, and for drawing our attention to the missing comparisons and analyses from our submission. We address your comments below.
>
>
> 1. Scope and novelty of justification for vulnerable code
>
>
> We agree that the use of natural language reasoning for detecting vulnerabilities is not a novel contribution; existing work such as VulLLM and R2Vul has utilized rationales in a structurally similar way in which natural language justifications are used. Our contribution supports this reasoning in two aspects that are complementary: (a) ARGO employs a composite label-free grounding reward that is computed from the code and output from the model (i.e., no CWE labels, no external teacher, no retrieval) and (b) both the composite reward and calibrated with a single probability with a deterministic threshold, human-calibrating both aspects of detection quality and calibration are core contributions. We will clarify your suggestion in the introduction that we are contributing a rationales that is composite reinforcement and calibrated around detection and classification, not just the use of rationales.
>
>
> 2. Choice and strength of baselines (including R2Vul)
>
>
> We acknowledge that R2Vul [1] and other calibrated / RL methods are appropriate baselines. While we will consider this as a reasonable issuing, we initially excluded them for two practical reasons that we did not state: (i) we wanted methods to run under the same token and epoch budgets on the three datasets, and (ii) the recent system (including R2Vul) considered variable model families, dataset mixes, and supervision signals throughout each paper, and thus the apples-to-apples comparisons are not strictly relevant.We intend to enhance the baselines in two ways:  • Add one strong RL-from rationales baseline at a minimum, using the public R2Vul code and configuration to the extent feasible in comparison to our training budget; if retraining with all datasets is impossible in the review window, we will at least include the released R2Vul checkpoints as baselines in instances where they support some portion of the DIVERSEVUL / BIGVUL / CLEANVUL tasks.  • Extend Sec 2 and Sec 5 to explicitly position ARGO against R2Vul and VulLLM: specifically, R2Vul applies RLAIF with preference data distilled from a strong teacher, while ARGO relies on self-computed grounding signals and a well-formed scoring term without external labels for preference;
>
> 3. Ablation and design justification
>
>
> We agree the main text could better position ablations, in the submitted version ablations for each component (of removing grounding rewards, removing formatting checks, turning of the surprise gained advantage modulation, reducing ARGO to pure SFT) were shared in the appendix and indicated that:  • Removing the explanation grounding mostly only harms benign recall and calibration, and leaves vulnerable recall near that of the baseline SFT (more to report here about how these terms change calibration as it relates to CCE);  • The removal of formatting checks does result in a moreWe will shift the most informative rows of these tables into the main text and add one more experiment that isolates the effect of the two-phase structure (forcing the model to output only a label vs. rationale-plus-probability with the same training budget).
>
> 4. Dataset granularity and CWE coverage
>
> All three of the benchmarks we use are labeled at a snippet or function level, and CLEANVUL is function-level specifically. Our design goal is to remain faithful to this standard setting, and study to what extent calibration and benign recall performance can be improved without altering the labeling granularity. We will add a brief mention in Sec. 4, acknowledging that many real-world pipelines operate, and desires richer program contexts, and that ARGO's ideas could extend to path-level or project-level inputs as well.
> For CWE coverage, we do also agree that a per-CWE (or per-CWE-group) breakdown would be helpful to see and understand if changes are driven by just a few classes. When we have CWE metadata available in BIGVUL and DIVERSEVUL we will add tables in the appendix that report macro-F1 per coarse CWE family, and provide a brief overview of changes in the main text, for transparency, but without changing the substantive takeaway points.
>
> 5. Terminology around “unsupervised” format coherence reward
>
> Our intent was to emphasize that both the grounding and format rewards do not use vulnerability labels, CWE tags, or preference data - but are computed with raw code and the model’s own output only. Referring to them as “unsupervised with respect to vulnerability labels” was certainly more precise than simply “unsupervised. In revision, we will change the naming to “label-free grounding and format rewards” and add a sentence early in Sec 3.2 that explicitly states what supervision they do and do not use.

---

### Official Review · Reviewer_hGKc · 2025-11-01

**Soundness:** 1
**Presentation:** 1
**Contribution:** 1
**Rating:** 2
**Confidence:** 4

**Summary:**

This paper aims at using LLM to detect vulnerability. It proposes an RL training algorithm with a novel set of rewards.
While the paper is targeting at an important problem, I have significant concerns about its soundness and evaluation setups.

**Strengths:**

It targets an important problem with creative training objective design.

**Weaknesses:**

1. The experiment setup contradicts with consensus in the vulnerability detection domain. The experiment is conducted on individual functions. However, it is well established that the vulnerability of code snippet should consider coding contexts. It might not be meaningful to consider function separately when reasoning about its security.

2. The unsupervised reward design lacks rationale.
The paper proposes three reward signals for evaluating whether a natural language explanation about the give code is plausible. First, it evaluates how many variables in the code are mentioned in the explanation (more means better).
Second, it evaluates whether the sentence matches code statements. Actually, I cannot understand the reward clearly from the writing. At line 199, the function S(x) and function match(u, s) were not defined. I don't know what those functions mean.
Third, it evaluates whether the reasoning mentions "Issue", "Evidence", and "Mitigation" sections in the explanation.

From my understanding, the rationale behind those reward signals are unclear. For example, it is unclear why an explanation that mentions more variables is a better explanation about certain vulnerability. Moreover, the reward design seems very vulnerable to reward hacking. An explanation that simply repeats the code line by line will result a high reward. It is not related to the final task (i.e., vulnerability detection) at all.

The unclear rationale behind reward design significantly undermines the paper's technical contribution.

3. The RL algorithm lacks justification.
The paper proposes novel training losses to compute the advantages. It proposes a novel term named "surprise statistic". However, there is no ablation study to justify the effectiveness and improvement of the new algorithm. The contribution of such algorithm is unclear.

**Questions:**

Please see the above discussion.

---

> ### Author Response · Authors · 2025-11-15
>
> We thank Reviewer hGKc for the thoughtful comments and for highlighting places where the current draft is unclear. We address the three main concerns below and will revise the paper accordingly.
>
> 1. Experimental setup and code context
>
> We agree that real-world vulnerability assessment often needs broader program context (inter-procedural flows, configs, build flags). Our goal in this work is narrower: to improve robustness and calibration in the standard snippet or function-level setting used by the public benchmarks we study (DIVERSEVUL, BIGVUL, CLEANVUL), which are themselves labeled at that granularity. ARGO does not assume that vulnerabilities are truly local; it assumes only that these datasets define a widely used proxy task. We will make this scope explicit in the introduction and Sec. 4 and add a short discussion of how the same composite-reward and two-phase formulation can be applied to longer contexts in future work.
>
> 2. Rationale and robustness of the unsupervised reward
>
> We apologize for the notational gap: S(x) and match(u, s) should be defined in Sec. 3.2. In words, S(x) is the set of normalized statement spans extracted from the snippet, and match(u, s) checks whether an explanation sentence u refers to a statement s after normalization (identifier canonicalization, whitespace and comment stripping). We will add explicit definitions and a brief pseudocode description.
>
> The three explanation terms are designed to encourage grounded, auditable reasoning without relying on CWE supervision:
>
> • Identifier coverage: encourages the model to mention the concrete program entities it is basing its decision on, rather than generic templates. The term is normalized by the number of distinct identifiers so it quickly saturates; repeating names does not keep increasing reward.
>
> • Sentence–code alignment: encourages each sentence, especially in the Evidence section, to be anchored to at least one code statement in S(x). Generic text that does not refer to operations in the snippet therefore receives low alignment.
>
> • Structure (Issue, Evidence, Mitigation): reflects how findings are presented in practice and is a simple bounded check for section presence. Its weight is small compared to the supervised decision term.
>
> Crucially, the composite reward always contains a clipped proper scoring term on the label, weighted more heavily than the unsupervised parts. A policy that simply echoes code line by line but predicts an incorrect label is penalized by this term and does not obtain high total reward. In Sec. 5.2 we already summarize ablations (full tables in the supplement) showing that removing explanation grounding mainly hurts benign recall while leaving vulnerable recall similar, indicating that these signals are in fact correlated with better task performance rather than disconnected from vulnerability detection. We will move a compact ablation table into the main paper and expand Sec. 3.2 to clearly explain how coverage, alignment, and schema validation jointly discourage trivial reward hacking strategies.
>
> 3. RL algorithm and “surprise statistic”
>
> Our RL update is intentionally conservative: it is standard single-sample policy gradient with a batch baseline. The “surprise statistic” is a nonnegative scalar computed from how atypical an explanation looks relative to its batch neighbors; it only appears inside a bounded, monotone modulator psi(s_j) that rescales the advantage and is constrained to a fixed interval. We do not claim any special optimality for this construction (and state this in Sec. 3.4); its purpose is to modestly reweight rare but informative episodes while keeping the estimator stable.
>
> The empirical evidence for the usefulness of the RL algorithm already appears in Sec. 5.1–5.2: under matched token budgets, ARGO consistently outperforms supervised fine-tuning (and zero-shot baselines) on macro-F1 and calibration across all three datasets. That is, the algorithm as a whole is evaluated against a strong supervised baseline. We agree that an explicit ablation of the surprise modulation itself (for example, setting psi to a constant so the estimator reduces to a pure neighborhood-smoothed baseline) would further clarify its role, and we will add such a comparison in the appendix if space allows, making clear that the main contribution lies in the composite reward and two-phase formulation rather than in this specific modulation.
>
> We hope these clarifications address the concerns about soundness and evaluation, and we will revise the paper to improve the definitions, motivation, and stated scope accordingly.

---

### Meta-Review · Area_Chair_w26N · 2026-01-07

**Summary:**

This submission addresses an important problem—improving calibration and explainability in LLM-based vulnerability detection—by proposing a composite reinforcement fine-tuning framework. Reviewers appreciate the motivation and some empirical improvements over supervised fine-tuning. However, they raise substantial concerns regarding the soundness and justification of the proposed reward design, the novelty and clarity of the reinforcement learning components, limited and insufficiently strong baselines, and the reliance on snippet/function-level evaluation that weakly reflects real-world security reasoning. Poor presentation and unclear exposition further undermine confidence in the contribution. Taken together, these issues motivate a recommendation to reject.

**Reviewer Concerns:**

The rebuttal partially clarifies notation, scope, and intended motivation, and acknowledges several limitations (e.g., missing baselines, need for clearer ablations, terminology around “unsupervised” rewards). However, the core concerns remain largely outstanding. In particular, reviewers remain unconvinced that the label-free grounding rewards meaningfully capture correct security reasoning or are robust against reward hacking, and the connection between these signals and vulnerability understanding is still weak. The RL algorithmic components lack strong theoretical or empirical justification, with key ablations not sufficiently emphasized in the main paper. Evaluation remains limited in scope, with missing comparisons to recent and relevant methods, and the presentation issues are significant enough that the technical contribution is difficult to assess reliably.

**Reviewer Scores:**

Given the rebuttal, reviewers’ scores would likely not change materially. Reviewers who rated the paper as reject or marginally below threshold would likely maintain their assessments, as the main concerns about soundness, evaluation strength, and clarity were not fully resolved during discussion.

---

### Decision · Program_Chairs · 2026-01-26

Reject